# A Review on Equine Influenza from a Human Influenza Perspective

**DOI:** 10.3390/v14061312

**Published:** 2022-06-15

**Authors:** Fleur Whitlock, Pablo R. Murcia, J. Richard Newton

**Affiliations:** 1Medical Research Council, University of Glasgow Centre for Virus Research, Garscube Estate, Glasgow G61 1QH, UK; fleurwhitlock1@gmail.com (F.W.); pablo.murcia@glasgow.ac.uk (P.R.M.); 2Equine Infectious Disease Surveillance (EIDS), Department of Veterinary Medicine, University of Cambridge, Madingley Road, Cambridge CB3 0ES, UK

**Keywords:** influenza, horses, equine, human, epidemiology, vaccination

## Abstract

Influenza A viruses (IAVs) have a main natural reservoir in wild birds. IAVs are highly contagious, continually evolve, and have a wide host range that includes various mammalian species including horses, pigs, and humans. Furthering our understanding of host-pathogen interactions and cross-species transmissions is therefore essential. This review focuses on what is known regarding equine influenza virus (EIV) virology, pathogenesis, immune responses, clinical aspects, epidemiology (including factors contributing to local, national, and international transmission), surveillance, and preventive measures such as vaccines. We compare EIV and human influenza viruses and discuss parallels that can be drawn between them. We highlight differences in evolutionary rates between EIV and human IAVs, their impact on antigenic drift, and vaccine strain updates. We also describe the approaches used for the control of equine influenza (EI), which originated from those used in the human field, including surveillance networks and virological analysis methods. Finally, as vaccination in both species remains the cornerstone of disease mitigation, vaccine technologies and vaccination strategies against influenza in horses and humans are compared and discussed.

## 1. Introduction

Influenza A viruses (IAVs) are capable of causing disease in a variety of hosts including humans, equines, canines, felines, avians, sea mammals, bats, and swine [1,2,3,4,5,6]. Although influenza viruses were first isolated in 1933, accounts of probable occurrences were described as early as 412 BC, with an influenza-like disease reported by Hippocrates [7,8]. IAV is an enveloped virus with a segmented genome comprised of eight genomic segments of single-strand RNA and are designated a subtype, determined by the surface proteins, haemagglutinin (HA), and neuraminidase (NA) [9]. Similar to other RNA viruses, IAVs evolve rapidly, and mutations in the hemagglutinin (HA) and neuraminidase (NA) genes can lead to antigenic changes via antigenic drift [10]. Co-infections by different IAVs can result in reassortment due to the segmented nature of the viral genome, and antigenic shift is the result of reassortment that involves the genomic segments that encode for HA and/or NA. IAVs are thought to have originated in waterbirds, and they remain the natural reservoir. The origins of equine influenza virus (EIV) are believed to be the result of a direct spill-over event from birds [11]. In comparison, while human influenza viruses can carry genes of avian origin, their emergence usually involves the pig as an intermediate host to enable the transmission of avian IAVs to humans [12]. IAVs have caused disease of varying degrees and severities in humans all around the world, and equine influenza (EI) remains one of the most important respiratory diseases of horses in the 21st century. EI affects equines in most countries, with only Iceland and New Zealand known to have remained free from disease [13]. There are many similarities between human and equine influenza, including being highly contagious, causing respiratory disease [14,15], and the involving the possibility for widespread transmission as a result of the aerosolized virus and integrated contact networks, including inter-continental air travel, that are applicable to both species [16,17,18,19]. When these factors are combined with the virus’s ability to acquire antigenic changes, disease occurrences are common, even in vaccinated individuals. EI remains endemic in many countries around the world, and seasonal human influenza epidemics occur annually [20,21]. Influenza epidemics and pandemics also affect both species, usually following the introduction of novel IAVs. Ever changing, their threat is constant, and furthering our understanding of host-pathogen interaction is integral to inform control and prevention measures [22]. Monitoring and surveillance also assist in determining management options, and given the zoonotic potential of IAVs, an interdisciplinary approach in all species is essential [23]. This review will discuss the similarities and differences between human influenza and EI and will focus on the pathogenesis, immune responses, clinical aspects, and epidemiology including transmission, cross-species transmission threats, surveillance, and preventive measures such as vaccines [24].

## 2. Pathogenesis, Immune Responses, and Clinical Aspects

The pathogenesis of influenza in each species is similar and both viral and host factors determine the manner of disease development However, these aspects are still not fully understood in either species [25]. HA has a crucial role in IAV’s causing disease by binding to host respiratory epithelial cell surface receptors containing sialic acid to enter the cell and infect it [26]. The distribution of these sialic acid receptors within the respiratory tract is different in horses compared to humans [27]. The affinity of IAVs to the receptor is also species specific, with avian and equine IAVs binding to the sialic acid-α2,3-galactose linkage and human IAVs binding to sialic acid-α2,6-galactose [28]. Influenza virus tissue tropism also determines the virus’s ability to infect cells and the range of species an IAV strain can infect, with some strains having a greater ability to infect or a preference for particular body tissues, compared to others [29,30]. As such, strain specific tissue tropism has been shown to determine the resultant clinical signs encountered [25]. In addition to HA, it is known that other viral factors are also responsible for pathogenesis in both species. As an example, rapid viral replication in cells disrupts cell function resulting in damaged respiratory tract epithelium, a localized inflammatory response and the wider systemic toxic symptoms that occur in both species [26,31]. 

Host factors comprise an individual’s innate and adaptive immune responses. Physical barriers in both species are similar and include mucus and non-specific cellular responses, with IAVs able to impair these responses through damage to respiratory epithelia, reduced mucociliary clearance capabilities, and dysfunction in immune cells [32]. Subsequent adaptive immune responses involve cell mediated immunity including CD8+ cytotoxic T cells and CD4+ helper T cells and humoral immunity involving HA specific B cells that produce circulating and local production of antibodies for neutralization of the virus [33,34]. In contrast to humans, in equines, dams do not transfer maternal antibodies across the placenta to a fetus, with neonates therefore born immunologically naïve and colostrum being a vital source of immunity [35]. Host immune responses vary and can depend on factors such as age-related changes, with increasing age in humans and equines demonstrated to cause a reduction in the effectiveness of aspects of the immune response [36,37]. Consequently, age has also been shown to influence the immune response to influenza vaccines, resulting in a reduction in efficacy of influenza vaccines in that sub-population [38,39]. Studies in equines also demonstrate that exercise stress can suppress the immune response to influenza virus infection, with a subsequent increased susceptibility to disease [40]. However, there is currently little evidence available on the effects of exercise stress and susceptibility to respiratory infections in humans [41]. 

Immune dysregulation through an over exuberant immune response has been shown to be disadvantageous in both species. In humans, lung damage and severe disease due to immune dysregulation can occur following infection with particular influenza strains and/or high viral loads [25]. A fatal suspected influenza-associated encephalopathy was reported in an EIV positive horse, and this was hypothesised to be the result of an idiosyncratic immune response, and a similar condition was also reported in humans, with the pathogenesis not fully determined but both immune dysregulation and viral infections of neurons are hypothesized to cause the condition [42,43]. 

In both horses and humans, prior exposure to an IAV through infection or vaccination likely provides cross-protective antibodies to antigenically related strains, lessening disease [44,45,46]. However, this may not always be the case, and notable outbreaks, epidemics, and pandemics have occurred in populations despite a level of prior exposure or vaccination to other strains or subtypes [17,47,48]. In both species, the presence of such antibodies was demonstrated to exert immunological pressure on the virus to adapt through antigenic drift in order to evade the immune response [49]. To monitor for viral changes which may in turn impact the effectiveness of current vaccines, veterinary surgeons are actively encouraged to sample vaccinated equine cases with signs suspicious of EI [50]. Human recovery from influenza symptoms is reported to be around a week, but a cough can last two or more weeks [21]. Resolution of pathology such as epithelial damage to the respiratory tract in equine cases has been demonstrated to take a minimum of three weeks [51,52]. Optimal respiratory function is essential as competition equines are required for their athletic performance, and EI has the potential to have huge implications on their intermediate use as a result of the pathology and clinical disease it can cause. 

Furthering our understanding of influenza pathogenesis and of the unique interaction between virus and host is integral to controlling and preventing disease in the future. With this in mind, the horse could be argued to be a good animal model for studying human influenza pathogenesis [34]. 

Human influenza virus infection may be indistinguishable from other common respiratory viral infections, and without additional diagnostic testing, many occurrences will be undiagnosed [53] and this is thought also to occur in the equine field. In both species, the resultant disease is most commonly an acute respiratory infection of varying severity, with clinical signs of pyrexia, coughing, nasal discharge, conjunctivitis, and lethargy [15,21]. Additionally, humans report a general feeling of malaise, muscle and joint pain, sore throat, and headache, with these all being symptoms that could also be experienced by a horse but are not clinically identifiable. In both species, disease severity varies by individual, determined by host factors such as the level of immunity from previous exposure or vaccination, co-morbidities, and viral factors such as virulence and infectious dose [54,55,56,57]. Rapid diagnosis in both species is essential to achieve optimal outcomes. In cases of human influenza, a quick diagnosis assists in determining optimal treatments [58,59] and in horses, control measures can be implemented to limit the onward spread of the virus. Following infection and peak viral shedding, clinical recovery usually occurs, or other more severe sequelae are possible, and, in both species, disease is occasionally fatal. EI can cause a high level of disruption through loss of horsepower for work and daily tasks, cancellation of equine competitions and leisure activities, treatment and control costs, and not forgetting the animal welfare implications from clinical disease. An epidemic in a naïve horse population in Australia in 2007 was estimated to have cost A$1 billion [60]. Seasonal epidemics of human influenza have also been shown to have a substantial socio-economic cost, with an estimated average cost of $11.2 billion in the USA in 2015 [61]. However, exact figures are not always attainable, particularly for low- and middle-income countries (LMIC) [62]. Human case fatality rates vary widely, as is also thought to be the case with EI, and data are biased and suffer from underreporting [63]. In the 2019 EI epidemic, a highly susceptible population of donkeys in West Africa was reported to have 60,000 deaths [64]. Deaths have also been reported in other specific EI epidemics and rarely in neonatal foals [65,66]. The human influenza pandemic in 1918 was responsible for 20 million deaths worldwide [67]; the USA documented between 12,000 and 52,000 deaths annually between 2010 and 2020, and an estimated 500,000 deaths worldwide are attributed to influenza each year [68,69]. At-risk human populations for severe sequelae of death from seasonal influenza are the elderly; however, during pandemics, there is often a high mortality in younger people [70]. Other risk factors associated with poorer outcomes following human influenza infection include a prior diagnosis of asthma, obesity, and pregnancy [58,71,72] but risk factors for complications in equines following EI are not widely studied. Genetic predispositions in humans have also been explored [73] and again, this is an understudied area in equine research, but it is clinically reported that certain species and breeds are prone to more severe disease following infection, such as donkeys [64,74,75]. It is suspected that unexpected sequalae from EI may go undiagnosed or unreported due to no veterinary involvement, a lack of follow-up testing such as post-mortem examination, or missed reporting opportunities by veterinarians. The sequela of pneumonia can occur in both species and can be primary viral or secondary bacterial [76,77,78,79,80]. A diffuse viral pneumonitis causing hypoxemia and a viral-induced acute respiratory distress syndrome can occur in humans and has been reported in horses [66,81]. Sporadic occurrences of atypical manifestations in both species include myositis, myocarditis, and encephalitis [43,82,83]. Guillain-Barre syndrome is also a human-reported complication, although no specific reports in horses exist. 

## 3. Epidemiology including Factors Contributing to Local, National, and International Transmission

The only EIV subtype currently circulating is H3N8, first isolated in 1963 [84]. Previously, H7N7, first isolated in 1956, did circulate but has not been identified since the 1980s [85,86]. The circulating EIVs are further classified into sublineages and strains. In both species, viral circulation in different geographical areas has resulted in different IAV strains. Classically, Florida Clade 1 strains were thought to circulate in North America and Florida Clade 2 in Asia and Europe; however, Florida Clade 1 has become the only strain found to be circulating in Europe since 2019, with Florida Clade 2 only being isolated on one occasion worldwide since 2019 [87,88,89,90,91,92]. International movement of horses and subsequent poorly implemented quarantine measures in Europe could be responsible for the recent changes seen in the geographical location of Florida Clade 1. The evolutionary dynamics of EIV described here were compared to those seen with influenza B virus in humans and are thought to be similar [93]. 

Currently circulating seasonal human influenza subtypes are H1N1 and H3N2. The current H1N1 strain was first isolated during the 2009 pandemic, and this virus strain replaced the previous human seasonal H1N1 [94]. H3N2 human IAV was first isolated during the 1968 pandemic and since then has caused seasonal influenza illness and death. While both H1N1 and H3N2 evade immune responses, the rate of antigenic drift of H3N2 is much higher than H1N1 [95]. Furthermore, H3N2 was found to cause a higher disease burden in the elderly, being responsible for more hospital admissions and deaths than H1N1 [96]. There are additional influenza subtypes of concern with pandemic potential including H2N2 (which has been absent for 50 years from the human population but could re-emerge from its waterbird reservoir), as well as H5N1 and/or H7N9 if they acquired the ability to transmit between humans effectively [97,98]. It is not fully known which novel subtypes could be of a concern in equines. Previous studies show that avian H3N8 viruses, distinct from equine H3N8 viruses, were isolated in outbreaks of equine respiratory disease, and equine exposure was also confirmed through serological surveys [84,99,100]. Additional strains of concern could include H7N7, as, if it re-emerged, given its proven ability to transmit between horses and a worldwide naïve population, its effects would be widespread. In addition, H5N1 remains a strain of concern, having been previously isolated from donkeys during an outbreak [101]. Finally, an immune response to canine H3N2 was previously identified in horses through serology, demonstrating that it may also have the ability to jump species [102].

Although antigenic drift of EIV commonly occurs, there is no evidence of novel EIVs as a result of antigenic shift [103,104]. Antigenic drift of EIVs is evaluated through antigenic characterization, and antigenic properties of EIVs are analyzed using antigenic cartography, with this technique being initially developed for human IAVs [105]. More recently, the increasing use of next generation sequencing improved our understanding of IAVs’ genetic and antigenic evolution [106]. Such analyses in one study demonstrated that EIV H3N8 was grouped into three distinct antigenic clusters, and clusters that developed from 1968 to 2007 were demonstrated to have been caused by specific amino acid substitutions [103]. The temporal development of each cluster was also identified, with one appearing 21 years after the first and the third appearing 14 years after the second [103]. When these data are compared to human IAVs over the same time period, there were 11 antigenic clusters, highlighting the increased diversity between human strains in comparison to equine [105]. The rate of antigenic drift was also previously quantified for IAVs, with the average rate of H3 EIV found to be 2.1 nucleotide substitutions per year in one study, with additional studies also finding EIV to have a consistently low substitution rate [11]. In contrast, human influenza H3 virus’s average rate was reported to be over double that of EIV at 5.9 nucleotide substitutions per year [49,103]. It is not clear why EIV has this lower rate of evolution [103].

Transmission of influenza in humans and equines is possible by three different routes, direct transfer of droplet particles, aerosols (droplet nuclei), and indirect means such as contaminated hands or objects [107]. In both species, coughing and, although less common in equines, sneezing produce droplet particles that can be transmitted in contacts. Aerosol transmission in equines, in certain conditions, has enabled EIV to spread distances of up to one and a half kilometers [108]. Viral, host, and environmental factors all determine transmission dynamics. The common sources of EIV include new arrivals to equine premises, temporary mixing of equines from different populations such as at equine competitions, international travel, personnel moving between different equine populations, and contaminated equipment (Figure 1). Transboundary transmission is confirmed as a cause of many EI outbreaks as a result of international transport of horses for activities such as equine competitions and breeding [19,109]. Horses traveling internationally are most likely competing under rules of a regulatory body that imposes influenza vaccination legislation requirements, or receiving countries may have pre-entry vaccination requirements, so they should be vaccinated. However, vaccinated horses may still shed the virus but may not have overt clinical signs, resulting in a lack of awareness for the requirement of control measures and onward disease transmission. Biosecurity measures, in addition to vaccination, to combat infection have been found to be ineffectively implemented in equine populations, such as a lack of quarantine of new arrivals to a population, limited hygiene practices such as handwashing between handling different groups of horses, and insufficient vaccine coverage of populations [90,110,111,112]. The EI outbreak in Australia in 2007 was associated with the movement of stallions for the southern hemisphere breeding season, with EIV introduced to the naïve resident horse population following a breakdown in quarantine biosecurity [113].

The common sources of human influenza infection include respiratory droplet transmission following a close contact sneezing and fomite transmission on contaminated surfaces. Measures to avoid infection are usually focused around encouraging sick persons to stay at home and personal protection measures such as avoiding close contact with sick people and hand washing [114].

Following exposure in both humans and equines, the incubation period is usually around one to two days, and the virus may be shed for around four to seven days, with a peak of shedding around four days after exposure and resolution of infection at day ten [54,115]. More prolonged shedding is seen in children due to a lack of immunity, and specific studies in horses have not compared viral shedding by age, but younger horses are generally more susceptible to infection [116,117]. The most significant modes of transmission in humans are still debated, particularly about whether transmission requires close contact [118]. Establishing this is essential for determining optimal control measures to limit spread, with measures for preventing airborne transmission being very different from those preventing droplet and contact transmission. Additionally, different influenza strains may be adapted to spread by one particular transmission pathway more effectively, and the host and environment will also impact this [119]. Experimental challenge studies in equines demonstrated that, although viral loads were similar, clinical sign severity and duration could be different depending on the infecting viral strain, with this consequently leading to variations in transmissibility, independent of viral load [120]. An important question is whether human influenza can travel long distances, and studies demonstrate that this is unlikely as viral particle sizes produced from coughing are mostly too large. Human IAV strains have also been shown to decay at a faster rate compared to animal strains [121,122]. Studies show that human transmission is most commonly a result of short-distance transmission through droplets or contact [110,111,118].

## 4. Cross-Species Transmission Threats

Horses have been domesticated by humans for millennia; however, the human need for horses for daily tasks has diminished hugely and, as a result, so have the numbers of horses [123,124]. However, equids are still vital for the daily lives of many people living in LMICs, and they may live in close confines with humans. Horses kept for leisure purposes will also be in close proximity with humans, with many kept in stables with relatively closed airspaces and requiring daily care. With animals being responsible for almost two thirds of emerging human infectious disease, and given that all identified human pandemic IAV strains since the 1800s have originated from an animal host, understanding influenza transmission threats between horses and humans is essential to ensure rapid detection if and when they do occur [125]. In most circumstances, cross-species transmission of IAVs usually results in limited onward transmission in the new host, and although EI infections among humans with horse exposure or through experimental infection have been confirmed serologically, human-to-human transmission has not been reported [67,126,127,128,129]. As there are significant differences between EIV and human influenza virus sialic acid receptors, there remains a host range restriction for IAVs [130]. Additionally, innate immune responses in the host prevent viral infection, although IAVs have evolved strategies to overcome such barriers [131]. It was also suggested that to support onward transmission, the new host population would need to be large, with integrated contact networks [1]. Such population structures exist in the human population and livestock agricultural sectors. However, other viral and host genes are also responsible for determining host range and need exploring to quantify and monitor transmission threats fully [131]. Interspecies transmission of H3N8 EIV occurred in canines in the late 1990s, with sustained circulation in the canine population in America, and sporadic spill-overs in the United Kingdom and Australia were reported [1,132,133,134,135,136]. Equids therefore have been an intermediate host for cross-species transmission of IAV. Furthermore, H3N8 EIV has been isolated from a camel, pigs, and infected cats under experimental conditions (Figure 2) [137,138,139]. 

## 5. Surveillance

Given the ever-changing nature of influenza viruses, surveillance of epidemiological and virological data is essential to understand the severity and extent of occurrences and the likely protection from prior exposure or vaccination depending on the causative strain. This information is used to optimize control and prevention measures in both species. The requirement for influenza surveillance in humans on an international level has been identified for decades given IAVs’ ease of spread, and the World Health Organization (WHO) Global Influenza Surveillance Network (GISRS) has been conducting surveillance since 1952 [140]. The main aim of GISRS is to provide a global overview of influenza. Types and intensities of human influenza surveillance will be country-specific and may utilize data from clinical surveillance and virological analysis. Additional data collected may include disease severity and mortality data, vaccination uptake, and coverage [141]. A key aim of WHO GISRS is to ensure international coordination of influenza research.

For EIV surveillance, the World Organisation for Animal Health (OIE) established a global Equine Influenza Surveillance Program in 1995 following EIV outbreaks due to vaccine and circulating strain mismatches in 1989 in Great Britain [142]. EIV surveillance initiatives are country-specific and will vary depending on resource, equine population, and disease status [143]. Many countries struggle to implement successful surveillance due to a variety of reasons including a lack of funding, low owner perception regarding the benefits of diagnostic testing, and a lack of commercial laboratory engagement with surveillance requirements, such as submitting samples to OIE reference laboratories for virus characterization [13]. EIV surveillance programs may include a combination of different elements of surveillance such as subsidized schemes to aid the diagnostic identification of disease, reporting systems to inform stakeholders of disease occurrence, and funded research programs for virological analysis.

For example, surveillance of EIV in Great Britain is conducted through an industry-funded scheme enabling subsidized laboratory fees to encourage testing of suspect cases by veterinary surgeons [144,145]. Epidemiological data collected may include population- and case-specific data to understand infection origins and spread and to optimize control and prevention measures. Virological analysis of positive isolates is conducted by OIE reference laboratories and by research institutes, and an annual bulletin is published by the OIE summarizing worldwide surveillance findings [91]. The International Collating Centre (ICC), an industry supported, GB-based surveillance reporting initiative, serves to inform stakeholders worldwide of voluntary reports of infectious disease occurrences such as influenza, and an influenza-specific platform for data interrogation, equiflunet, is also available [20,146]. GB-based equine veterinary surgeons can also sign up to a text-alert system, sponsored by a vaccine manufacturer, to receive real-time reports of confirmed influenza occurrences in GB [147]. The ICC obtains information on disease occurrences from country-specific reporting platforms such as France’s Réseau D’épidémio-Surveillance En Pathologie Équine (RESPE) and the USA’s Equine Disease Communication Center (EDCC) to name a few, and when an influenza report is shared with the OIE or if an influenza-free country confirms infection, this is reported through the World Animal Health Information System (WAHIS) [148].

Both human and equine influenza surveillance suffers from bias and underreporting. In a cohort study evaluating adults with acute respiratory illness, it was found that two thirds of patients with influenza were not diagnosed by clinicians [149]. During the 2019 EI epidemic in GB, it was suspected that a large number of outbreaks went undiagnosed and unreported [90]. Equine surveillance methods mainly rely on the detection of laboratory positive EI cases, and for a case to reach the reporting stage, a specific pathway must be completed (Figure 3), and multiple factors influence the uptake of each stage in this pathway, with these being unique to each country and their specific equine industry. An understanding of these factors is required and subsequent initiatives implemented to improve surveillance data quality and quantity [150,151,152].

Monitoring isolates during the seasonal human influenza period is essential to ensure that strain inclusion in vaccines is optimized. So called “immune-escape” variants as a result of antigenic drift may occur during a single epidemic, and research demonstrates that monitoring can be optimized if applied in the latter part of the human influenza season [153]. Surveillance in both species has similar challenges to overcome, and alternative strategies to diagnosing cases through laboratory testing include bioaerosol surveillance, which has been applied in both agricultural and human clinical settings for research purposes, and its benefits include that it is low-cost, non-invasive, and samples are more easily obtainable [154,155]. 

The first accurate report of a human pandemic was in 1580, and numerous reports of pandemics, with spread across the world, have been reported since. Most notable was the 1918 “Spanish flu” [156], the Asian pandemic of 1957 [157] and “Swine flu” in 2009 [47]. Countries with no confirmed outbreaks of EIV are New Zealand and Iceland, although other countries such as Australia and Japan have eradicated EIV. Notable epidemics affecting equids include “The Great Epizootic of 1872” in the United States, resulting in extensive disruption to human transportation and essential services [158], and more recently in 2007, EIV demonstrated its plague-like abilities spreading across the highly susceptible Australian equine population with over 10,000 infected [108]. Notable strain changes have occurred including in Hong Kong in 1992, when it was first identified that there were two lineages of H3N8 (American and Eurasian) and in the UK in 2003 when Florida Clade 2 was first identified [17,159]. In 2019, EIV was responsible for the halting of racing in Great Britain for only the second time in 40 years, and a parallel epidemic across West Africa was reported to involve over 60,000 donkey fatalities [64,90]. Disease was noted in vaccinated horses during 2019 across Europe, and reasons for this were debated. Vaccine and circulating strain mismatch was shown as unlikely from virological analysis, and in France it was speculated that poor population vaccine coverage and limited implementation of biosecurity measures may have been responsible [92]. 

## 6. Preventive Measures

Vaccination is the cornerstone of control and prevention of IAVs in both species, with vaccines administered with the intention to provide neutralizing antibodies and facilitate cell-mediated immunity. The majority of available vaccines in both species invoke the development of neutralizing antibodies by the host to HA, but a successful IAV vaccine would ideally invoke a robust, cell-mediated immune response too [160]. The first human influenza vaccines, available in the 1940s, were inactivated monovalent vaccines [161]. Equine vaccines were not available until the 1960s, and similarly, an inactivated vaccine was the first vaccine type and remained the main EI vaccine for decades. There have since been significant advances in EI vaccine technologies including the development of subunit vaccines adjuvanted with immune-stimulating complex (ISCOM) technology [162].

In both species, most currently available vaccines are still inactivated whole virus vaccines and are administered intramuscularly. A modified live-attenuated vaccine may be administered to children intranasally, and a similar intranasal vaccine also exists for equids [163,164]. Additional vaccine types may also include subunit, inactivated split virus, and recombinant virus vector, and their availability for each species and country is determined by country-specific licensing (Table 1). Additionally, DNA-based vaccines are currently undergoing research in both the human and equine field [165]. 

In humans, vaccines may contain the two currently circulating IAV strains and one or both influenza B strains [176]. Human vaccine strain selection is overseen by WHO and vaccines are updated regularly, to ensure they contain the most relevant influenza strains [177], with human vaccine manufacturing taking at least six months [178]. Equine vaccine strain inclusion is also overseen by an advisory group, the OIE Expert Surveillance Panel (ESP) on Equine Influenza Vaccine Composition, but vaccines are not updated in as timely a fashion, in part due to EIVs slower rate of antigenic drift [91] but also because of the time and cost associated with doing this for a relatively much more limited commercial market. Equine vaccines vary in their strain inclusion depending on the vaccine brand and vaccine type and the amount of time elapsed since an update was made by the manufacturer. As an example, only one currently commercially available EI vaccine in the UK aligns with the most recent OIE ESP guidelines. In the human population, specific applications of vaccine are country specific, with many targeting high risk groups such as the elderly and pregnant women, and children due to them acting as spreaders [117,179]. In equine populations, vaccination may be mandatory if a horse competes under rules of a regulatory body, such as racehorses in the United Kingdom [180] and vaccination six weeks prior to parturition is advised in pregnant mares, to enable offspring to obtain passive immunity from colostrum [181]. Commencement of vaccination in equines requires a primary course of two vaccines (excluding live attenuated EI vaccines) given between four and six weeks apart and then a booster around five months after the second dose. The vaccine is then licensed for annual boosters, however, studies have shown that there are huge benefits in decreasing this time frame to six months, particularly given the poor vaccine uptake in some countries and populations, the young age of many equines that commonly mix at equine competitions and the lack of updating of strains in vaccines to align with current circulating strains [182]. The effectiveness of current equine vaccines is monitored and currently available vaccines induce a measurable antibody response and these products are licensed to reduce clinical sign severity and duration, and viral shedding. It has been found that some equines will respond better to vaccination than others and no studies have been conducted to understand vaccine effectiveness in donkeys [183,184]. Studies have demonstrated that a greater level of immunity is obtained in horses that have had a greater number of vaccines during their lifetime and time since last vaccination is also important, with more protection from more recent vaccination [185]. Population medicine is essential in equines, with many living in large groups in close proximity and shared airspace and studies have demonstrated that ideally 70% of the population must be vaccinated to prevent transmission and if influenza field strain changes, population vaccine coverage may need to increase to 95% [186,187].

In both species, the hemagglutination-inhibition (HI) antibody titre is currently the only universally accepted immune correlate used for determining protection against influenza, despite its limitations [188]. Early studies into EIV demonstrated that high levels of antibody were required for protection and virological protection levels have since been quantified, with one study finding that pre-exposure serum radial haemolysis (SRH) levels of >140 mm^2^ resulted in no infection in the host [189,190]. In comparison, the threshold SRH value associated with a 50% protective titre in humans is >25 mm^2^ [191]. It has been stated that there is a need to explore the options of using additional correlates of protection to the HI antibody titre in order to improve the understanding of vaccine effectiveness in populations [192].

In addition to antibody levels, it has been demonstrated that there needs to be a relatively close homology between strains in current vaccines and circulating field viruses. Vaccine mismatch in both species occurs and in the human field this is most commonly following minor viral changes to circulating seasonal influenza due to a lag time between vaccine strain inclusion choice and production. In 2017–2018, vaccine effectiveness for the seasonal influenza season was estimated to be 36% overall [193]. In equines, failure of vaccine efficacy, sometimes referred to as vaccine breakdown, has been reported on numerous occasions and may be a result of vaccine and circulating strain mismatch or due to host or environmental factors [92,194,195,196]. In the event of new circulating subtypes affecting either species following antigenic shift, vaccines are most likely to provide minimal to no cross-protection. Human vaccination suffers from similar limitations as equine vaccines, including their variable efficacy in specific populations, such as the elderly. In contrast, human vaccines rarely contain adjuvants, due to rare adverse events from adjuvanted vaccines, however adjuvants have been incorporated in certain vaccines targeting the elderly in some countries, or during pandemics [197,198]. Ultimately, adjuvants will not address the mismatch between vaccine and circulating strains which is usually responsible for the poor protection imparted by human vaccines. Uptake of vaccination in both species can be severely affected by public perception of vaccination efficacy and safety, or an overall distrust in vaccines [199].

Monitoring HA through surveillance has been a routine practice for EIV, with HA being the primary target of currently available EIV vaccines. Human vaccine updates occur when there are four to five amino acid substitutions at a minimum of two antigenic sites [200]. For EIV, three antigenic units when characterized using ferret antisera, would likely result in vaccine breakdown, however, vaccine update recommendations are usually made when there are 4.7 antigenic unit differences [103]. Furthering the knowledge around IAV immunology would assist in more optimal vaccine design, such as targeting conserved epitopes. The ideal equine vaccine would be safe, invoke a protective immunity that is of adequate duration without the need for a primary course and frequent boostering and be robust in the face of antigenic drift and shift. Human vaccine technology research includes developing universal vaccines, avoiding the need for regular vaccine updates and efforts for their development are ongoing [201]. Equine vaccines would benefit from utilising new methods to speed up vaccination production or consider wider use of anti-virals, which is used as a first line of defence in human influenza pandemics when vaccines are not yet available.

As vaccination does not impart full immunity to IAVs in any species currently, additional measures for successful prevention and control of IAVs are necessary. In equines, prevention measures include having a premises biosecurity plan that all resident owners adhere too, including a vaccination policy for influenza, quarantining of new arrivals for the correct time period and using optimal hygiene protocols, and understanding and managing the risk posed by temporary horse movements to equine competitions [110,111,202]. Measures to avoid EI introduction following international movements are set by the OIE and for countries free of EI, include the requirement that horses arriving from endemic countries receive a booster within a specific time period of travel [203]. Pre- and post-movement diagnostic screening tests and quarantine may also be imposed. Education of owners to identify early signs of infection should achieve the ability to obtain a rapid and correct diagnosis of EI, in conjunction with a veterinary surgeon. Control measures should be implemented at first suspicion of infection and may include isolation of affected animals. However, given that many equine premises buildings share air space and residents have integrated contact networks, limiting spread of EIV on a premises is very challenging and focuses on avoiding spread beyond the premises may be more achievable. Booster vaccinating in contacts may also be applied, when deemed appropriate. Voluntary movement restrictions on infected premises should also be imposed and premises in the surrounding area should be made aware of the risk from aerosolised transmission of EIV. With most countries having endemic EI, it’s control of EI in such countries is unlikely to be under any legal jurisdictions, however some influenza-free countries do have legal mandates for its control and prevention in place. Education and public perception of EI will vary by country and may be dependent on the predominance of equid use and the relative socio-economic importance of the equine industry. The extent to which these measures are implemented and their success will therefore vary by country and by each specific equine population. The OIE ESP acts to inform and unite countries in their approaches to the control and prevention of EI, but engagement is variable by country. Human influenza control and prevention measures and relevant policy are also country specific and separate approaches are necessary for seasonal and pandemic influenza, with WHO overseeing the global perspective [21,204]. Prevention measures mainly centre around vaccination and surveillance to monitor the virus. Control measures focus on similar themes as EI and may include self-isolation, symptomatic screening, national and international movement restrictions, social distancing, public hygiene and disinfection, quarantine, targeted vaccination and public communications [205]. 

Non-pharmacological interventions (NPSs) including wearing masks and contact tracing that were implemented in many populations to control the spread of severe acute respiratory syndrome coronavirus 2 were found to effectively reduce influenza transmission and as a result, there was a decrease in incidence of influenza during that time [206]. Efforts for effective and reactive pandemic preparedness strategies in the human field are extensive and targeted, however, the equine field does not currently have any specific international or national action plans available in the public domain in the event of occurrences of pandemic EI [207,208].

## 7. Conclusions

Much of what is known about EI has followed from techniques and findings in the human field, and more can be gleaned to continue to improve approaches to EI. Pathogenesis, including the intricacies of viral-host interactions in both species, is still fundamental to explore in order to combat IAVs. Although much is known about human risk factors for infection and subsequent sequalae, data are still lacking on this in the equine field. Equine veterinary surgeons are encouraged to understand their role in improving our understanding of EIV by sampling suspect cases, reporting unexpected events following infection, and encouraging post-mortem examinations. The epidemiology of infection in both species is similar, but with more emphasis on the ability of EIV to travel large distances by aerosolization and human viruses mainly relying on close contact for transmission. Virus, host, and environmental factors all have a strong influence on the specifics of disease occurrences, including the clinical severity, extent of spread, and success of control and prevention measures. Surveillance in both species has similar challenges to overcome and includes improving engagement and addressing the disparity in data quantity and quality between countries. EIV strains do not seem to have the same diversity and rate of change as human influenza viruses, and as a result of this and a smaller commercial pharmaceutical market, equine vaccine strain updates are not employed in as timely a manner as they should be. Alongside a relatively poor coverage by vaccination and variable implementation of additional biosecurity measures, equine industries remain at high risk of EI occurrences. The main question to pose to the equine world would be to consider what the current preparedness for strains with pandemic potential is and how would prompt vaccine development and production be achieved. Although cross-species transmission of IAVs remains a concern, the relative human disease threat from EIVs seems to be low, but given the ever-changing nature of IAVs, open communication and a collaborative, joined-up approach between all parties are essential.

## Figures and Tables

**Figure 1 viruses-14-01312-f001:**
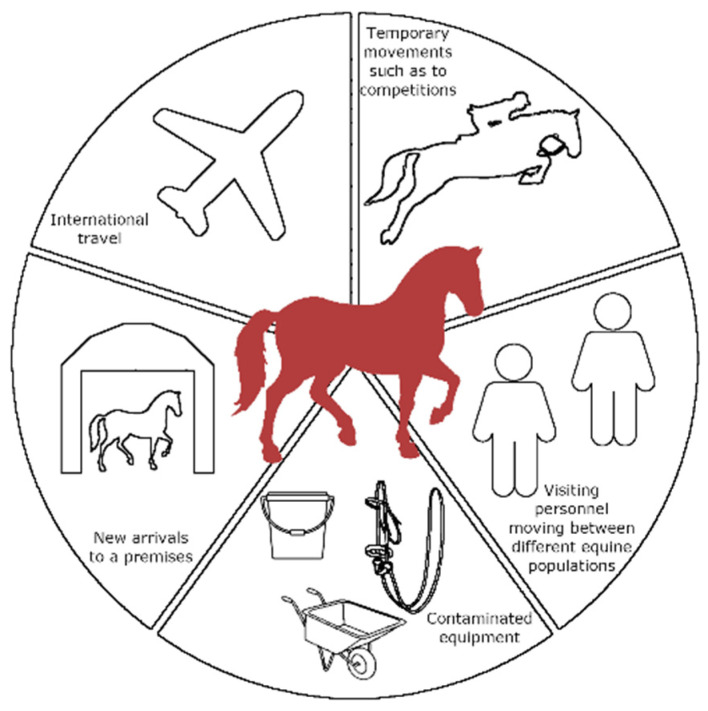
Common pathways for the introduction of an equine influenza virus infected horse to a population.

**Figure 2 viruses-14-01312-f002:**
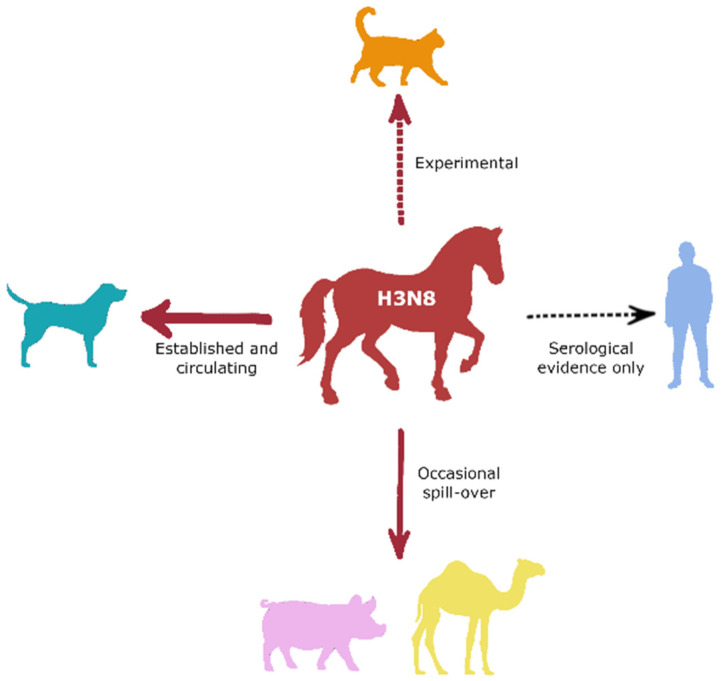
Cross-species transmission of equine influenza A virus H3N8. Thick solid red arrow represents direct transmission event that has since become established and now circulates within the species. Thin red arrow represents isolation from the species as likely spill-over events. Dashed red arrow represents isolation following experimental exposure only. Dashed black arrow represents serological evidence of exposure only, from field or experimental sources.

**Figure 3 viruses-14-01312-f003:**
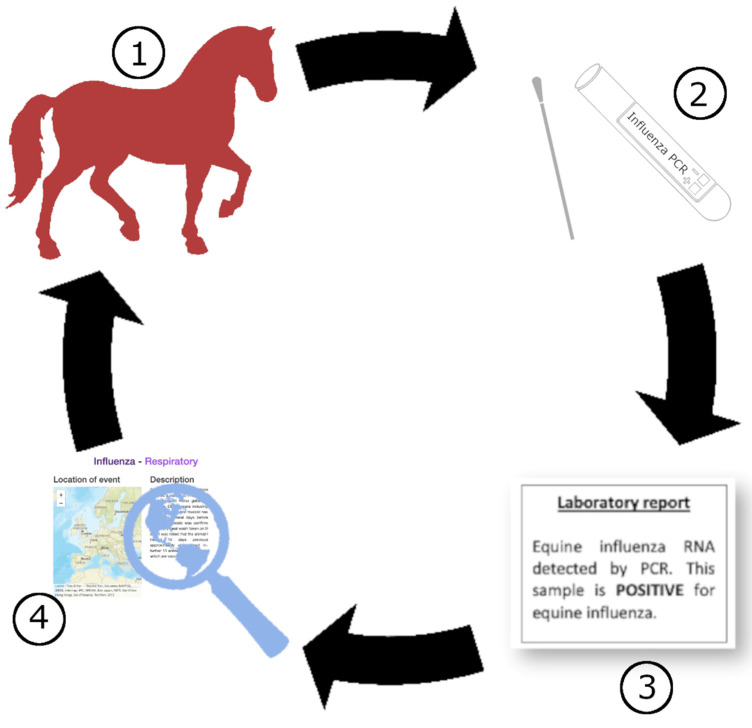
Equine influenza virus surveillance pathway for laboratory confirmed infection. 1. Infectious horse noted to be sick by equine keeper and veterinary surgeon contacted 2. Veterinary surgeon suspects infectious process, takes samples during infectious phase, and requests equine influenza agent detection testing such as polymerase chain reaction (PCR) 3. Laboratory correctly identifies equine influenza, and result is reported through a surveillance network 4. Surveillance network shares epidemiological data and analyzes viral isolate to determine influenza strain.

**Table 1 viruses-14-01312-t001:** A comparison of human and equine influenza commercially available vaccines.

Vaccine Technologies	Humans		Equines	
	Available	Particulars	Available	Particulars
Inactivated whole virus	Yes	Egg-based vaccine with excellent production capacity and low production cost but requires a supply of embryonated eggs and thus affected by shortages [166]. Contains complete virus antigenic components. Invokes a greater humoral immune response than a cellular one. Poorer cross-protection to viruses of different subtypes to that in the vaccine. Effectiveness in the elderly <50% [167]. High safety profile but not advisable for use in children as can cause a high fever. Used less now as subunit and split-virion vaccines are comparable immunologically and safer.	Yes	Requires multiple administrations to obtain a protective immune response and mainly invokes a humoral immune response [168]. Adjuvant added to improve immunogenicity [169]
Subunit	Yes	Produced using embryonated eggs. Safe vaccine [170]	Yes	Invokes a humoral and cell-mediated immune response [171]. Requires adjuvant to boost immunogenicity
Inactivated, split virus	Yes	Produced using embryonated eggs. Safe but poorer immunogenicity so requires two doses to overcome this.	No	Not available
Recombinant virus vector	Yes	Produced using virus vectors such as baculovirus (most successful) [172]. Higher cost of production than vaccines produced using eggs [166]. Induces cellular and humoral response and long duration of immunity.	Yes	Virus vectors used include canarypox. Immunity of a longer duration obtained. Invokes a robust cell-mediated response in addition to a humoral response. Has DIVA capability. Requires adjuvant to boost immunogenicity
Modified live-attenuated	Yes	Produced using embryonated eggs. As administered intranasally, produces local neutralizing antibody and a cell-mediated response [173]. Variable efficacy in adults but better efficacy in children [174]. Reversion to virulence or recombination with field virus is a possibility [175].	Yes	Invokes a long lasting, robust cell-mediated response in addition to a humoral response, without the requirement of adjuvants. Reversion to virulence or recombination with field virus is a possibility [162]

DIVA: differentiate infected from vaccinated animals.

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
