# Peer review of "A Review on Equine Influenza from a Human Influenza Perspective"

_viruses, 2022, doi:10.3390/v14061312_

Round 1

Reviewer 1 Report

Although generally well-written and quite thorough in some respects, I was quite disappointed with what claimed to be a ‘selective’ comparative review. I felt that the authors tried to cover too much ground and some interesting comparisons were missing, particularly as the bias of detail was towards equine influenza. I also felt that the manuscript would benefit from some restructuring to avoid unnecessary repetition (no need ot mention >once that the first isolation of an IAV was made in 1933). I would suggest cutting out a lot of detail to really focus on and highlight the comparative aspects. Some suggestions are made below.

Introduction

It could be argued there is no such thing as ‘equine influenza’ and ‘human influenza’ – only influenza A viruses, some of which have established circulation in equids and some in humans. The introduction would be better if it introduced influenza A viruses in general and the commonalities (including structure, genome and replication) that then do not need to be dealt in detail. In other words, I do not believe that ‘Virology’ should be a separate subheading but rather part of the introduction.

An example of factors that could be mentioned in the introduction but are repeated because equine and influenza are discussed separately (rather than comparatively):

For example, in line 57 ‘IAVs are thought to have originated in ‘waterbirds’, line 65 ‘EIV is thought to have originated in birds’, line 75 human IAVs have originated from avian and swine viruses…. Basically, aquatic birds are thought to be reservoir of majority of subtypes of IAV – receptor specificity means that viruses can readily transmit and become established in horses directly form aquatic birds but it is believed that the pig is usually required as an intermediate host for transmission of avian influenza to humans.

Pathogenesis and clinical aspects…this is, in my opinion, overly detailed. I would argue that there is a lot of similarity between how IAV infection manifests in both species…and this is an argument for the horse being a good model for studying human influenza. There is a large section on preventive measures (mainly vaccination) but the immune response is not discussed much – there are a couple of interesting papers that suggest that similarities between the equine and human immune responses mean that the horse is a good model, e.g.

Horohov DW. The equine immune responses to infectious and allergic disease: a model for humans? Mol Immunol. 2015 Jul;66(1):89-96. doi: 10.1016/j.molimm.2014.09.020. Epub 2014 Oct 22. PMID: 25457878.

Arguably, the discussion of the different lineages of equine influenza should come not under virology but under epidemiology – the poorly understood reasons for the slower evolutionary rate of equine influenza HA and the co-circulation of lineages vs IAV is quite a fascinating area. There are similarities between equine IAV and influenza B virus in terms of co-evolution of lineages…it may just be a different mechanism of immune escape – instead of linear evolution, different lineages cycle. A mathematical modelling paper by Katia Koelle makes interesting observations in this respect – including the potential impact of differences in quarantine measures leading to geographical separation of evolutionary lineages –

Koelle, K., Khatri, P., Kamradt, M., Kepler, T.B., 2010. A two-tiered model for simulating the ecological and evolutionary dynamics of rapidly evolving viruses, with an application to influenza. Journal of The Royal Society Interface 7, 1257-1274.

For a comparative review, I’d prefer to see more discussion of this sort of contract than lines 215 – 233, which is incredibly equine-specific. Similarly, the section on surveillance is not particularly ‘comparative’ it is just describing how the equine surveillance system emulates the human surveillance system and is very UK-centric (no mention of the fantastic The Réseau d'Epidémio-Surveillance en Pathologie Equine in France?).

Again, the section that is primarily on vaccines is very equine focussed. Furthermore, the comparison of available vaccines focusses on the ‘particulars’ of the vaccine. I think many people find it surprising to hear that vaccines against ‘equine influenza’ were introduced very soon after the introduction of human vaccines, but that there were several innovative platforms introduced for equine influenza (e.g. ISCOM vaccines etc.) while the human influenza vaccines remained unchanged for decades. That situation may well reverse now as the COVID-19 pandemic has paved the way for a step-change in vaccine platforms used preventively in the human population. However, it is still interesting to consider how the horse is a good model for determining the efficacy of different vaccines. Historically, data on vaccine efficacy/effectiveness has been difficult to obtain for human IAV vaccines.

Overall, I would strongly recommend that the authors consider editing their manuscript so that it is both more selective and comparative.

There are a few minor spelling and grammatical points, 

e.g. line 176 sequelae is incorrectly spelt

line 122 possessive apostrophes do not really work in scientific writing

There is some confusion around species as nouns and adjectives e.g. equine influenza (adj.) in equids (noun).

I think many would argue that there have been major epidemics of influenza in horses but not a pandemic

Author Response

Dear reviewer,

Many thanks for your time spent reviewing our manuscript. Please find below our point-by-point response.

Introduction

It could be argued there is no such thing as ‘equine influenza’ and ‘human influenza’ – only influenza A viruses, some of which have established circulation in equids and some in humans. The introduction would be better if it introduced influenza A viruses in general and the commonalities (including structure, genome and replication) that then do not need to be dealt in detail. In other words, I do not believe that ‘Virology’ should be a separate subheading but rather part of the introduction.

Reply: A general introduction to IAV has been included and reference to its origins only appear in this section now.

An example of factors that could be mentioned in the introduction but are repeated because equine and influenza are discussed separately (rather than comparatively):

For example, in line 57 ‘IAVs are thought to have originated in ‘waterbirds’, line 65 ‘EIV is thought to have originated in birds’, line 75 human IAVs have originated from avian and swine viruses…. Basically, aquatic birds are thought to be reservoir of majority of subtypes of IAV – receptor specificity means that viruses can readily transmit and become established in horses directly form aquatic birds but it is believed that the pig is usually required as an intermediate host for transmission of avian influenza to humans.

Reply: This suggested change has been made

Pathogenesis and clinical aspects…this is, in my opinion, overly detailed. I would argue that there is a lot of similarity between how IAV infection manifests in both species…and this is an argument for the horse being a good model for studying human influenza. There is a large section on preventive measures (mainly vaccination) but the immune response is not discussed much – there are a couple of interesting papers that suggest that similarities between the equine and human immune responses mean that the horse is a good model, e.g.

Horohov DW. The equine immune responses to infectious and allergic disease: a model for humans? Mol Immunol. 2015 Jul;66(1):89-96. doi: 10.1016/j.molimm.2014.09.020. Epub 2014 Oct 22. PMID: 25457878.

Reply: We have ensured that the pathogenesis section is more streamlined so to make clear the comparative aspects between human and equine flu pathogenesis. We have also included a section on immune responses as suggested.

Arguably, the discussion of the different lineages of equine influenza should come not under virology but under epidemiology – the poorly understood reasons for the slower evolutionary rate of equine influenza HA and the co-circulation of lineages vs IAV is quite a fascinating area. There are similarities between equine IAV and influenza B virus in terms of co-evolution of lineages…it may just be a different mechanism of immune escape – instead of linear evolution, different lineages cycle. A mathematical modelling paper by Katia Koelle makes interesting observations in this respect – including the potential impact of differences in quarantine measures leading to geographical separation of evolutionary lineages –

Koelle, K., Khatri, P., Kamradt, M., Kepler, T.B., 2010. A two-tiered model for simulating the ecological and evolutionary dynamics of rapidly evolving viruses, with an application to influenza. Journal of The Royal Society Interface 7, 1257-1274.

Reply: We have added a small section of what constitutes the biological basis of viral evolution of influenza in ‘Introduction’ and the molecular epidemiology of flu viruses described in ‘Epidemiology’. We have also added in the suggested reference.

For a comparative review, I’d prefer to see more discussion of this sort of contract than lines 215 – 233, which is incredibly equine-specific.

Reply: We have amended the manuscript in line with suggestions to ensure the focus was on similarities and differences between equine and human disease. If it is felt still exists beyond that of what is acceptable for this special review which looks to focus of equine and human viruses, we can consider changing the title so that the expectations of the reader match the content of the article.

Similarly, the section on surveillance is not particularly ‘comparative’ it is just describing how the equine surveillance system emulates the human surveillance system and is very UK-centric (no mention of the fantastic The Réseau d'Epidémio-Surveillance en Pathologie Equine in France?).

Reply: We have added in additional examples of surveillance systems.

Again, the section that is primarily on vaccines is very equine focussed. Furthermore, the comparison of available vaccines focusses on the ‘particulars’ of the vaccine. I think many people find it surprising to hear that vaccines against ‘equine influenza’ were introduced very soon after the introduction of human vaccines, but that there were several innovative platforms introduced for equine influenza (e.g. ISCOM vaccines etc.) while the human influenza vaccines remained unchanged for decades. That situation may well reverse now as the COVID-19 pandemic has paved the way for a step-change in vaccine platforms used preventively in the human population. However, it is still interesting to consider how the horse is a good model for determining the efficacy of different vaccines. Historically, data on vaccine efficacy/effectiveness has been difficult to obtain for human IAV vaccines.

Reply: We have included a brief description of the history of vaccines.

Overall, I would strongly recommend that the authors consider editing their manuscript so that it is both more selective and comparative.

There are a few minor spelling and grammatical points, 

e.g. line 176 sequelae is incorrectly spelt

Reply: This has been amended

line 122 possessive apostrophes do not really work in scientific writing

Reply: This has been removed

There is some confusion around species as nouns and adjectives e.g. equine influenza (adj.) in equids (noun).

Reply: ‘in equids’ has been amended to ‘in equines’, when appropriate

I think many would argue that there have been major epidemics of influenza in horses but not a pandemic

Reply: This has been amended

Reviewer 2 Report

This review gives overall summary of human and equine influenza A viruses. The authors have highlighted virology, pathogenesis, epidemiology, and transmission of human influenza viruses comparing with equine influenza A viruses. Overall, the manuscript is well written and provides comprehensive review of the subject matter. The way authors have compared vaccines for humans and equines, I recommend including images or tables in other sections including virology, pathogenesis and epidemiology as well. In pathogenesis section, specially, including images or further characteristics of pathological changes over time in equines is necessary. 

Author Response

Dear reviewer,

Many thanks for your time spent reviewing our manuscript. Please find our point-by-point response below.

This review gives overall summary of human and equine influenza A viruses. The authors have highlighted virology, pathogenesis, epidemiology, and transmission of human influenza viruses comparing with equine influenza A viruses. Overall, the manuscript is well written and provides comprehensive review of the subject matter. The way authors have compared vaccines for humans and equines, I recommend including images or tables in other sections including virology, pathogenesis and epidemiology as well. In pathogenesis section, specially, including images or further characteristics of pathological changes over time in equines is necessary. 

Reply: Tables have been included in a separate word doc for inclusion.